# 3,4-Methylenedioxypyrovalerone (MDPV) Sensing Based on Electropolymerized Molecularly Imprinted Polymers on Silver Nanoparticles and Carboxylated Multi-Walled Carbon Nanotubes

**DOI:** 10.3390/nano11020353

**Published:** 2021-02-01

**Authors:** Rosa A. S. Couto, Constantino Coelho, Bassim Mounssef, Sara F. de A. Morais, Camila D. Lima, Wallans T. P. dos Santos, Félix Carvalho, Cecília M. P. Rodrigues, Ataualpa A. C. Braga, Luís Moreira Gonçalves, M. Beatriz Quinaz

**Affiliations:** 1REQUIMTE, LAQV, Laboratory of Applied Chemistry, Department of Chemical Sciences, Faculty of Pharmacy, University of Porto, 4050-213 Porto, Portugal; rcouto@ff.up.pt (R.A.S.C.); tino44coelho@gmail.com (C.C.); 2Departamento de Química Fundamental, Instituto de Química, Universidade de São Paulo (USP), São Paulo, SP 05508-000, Brazil; bassim.mounssef@usp.br (B.M.J.); sara.famorais@usp.br (S.F.d.A.M.); ataualpa@iq.usp.br (A.A.C.B.); 3Departamento de Química, Universidade Federal dos Vales do Jequitinhonha e Mucuri, Diamantina, MG 39100-000, Brazil; camila.lima@ufvjm.edu.br; 4Departamento de Farmácia, Universidade Federal dos Vales do Jequitinhonha e Mucuri, Diamantina, MG 39100-000, Brazil; wallanst@ufvjm.edu.br; 5REQUIMTE, UCIBIO, Laboratory of Toxicology, Department of Biological Sciences, Faculty of Pharmacy, University of Porto, 4050-213 Porto, Portugal; felixdc@ff.up.pt; 6Research Institute for Medicines (iMed.ULisboa), Faculty of Pharmacy, Universidade de Lisboa, 1649-003 Lisbon, Portugal; cmprodrigues@ff.ulisboa.pt

**Keywords:** analytical chemistry, ‘bath salts’, biomimetics, drug analysis, electropolymerization, electroanalysis, forensic chemistry, modified electrodes, new psychoactive substances (NPS), *o*-phenylenediamine

## Abstract

3,4-methylenedioxypyrovalerone (MDPV) is a harmful and controlled synthetic cathinone used as a psychostimulant drug and as sport-enhancing substance. A sensor was developed for the direct analysis of MDPV by transducing its oxidation signal by means of an electropolymerized molecularly imprinted polymer (e-MIP) built in-situ on the screen-printed carbon electrode’s (SPCE) surface previously covered with multi-walled carbon nanotubes (MWCNTs) and silver nanoparticles (AgNPs). Benzene-1,2-diamine was used as the functional monomer while the analyte was used as the template monomer. Each step of the sensor’s development was studied by cyclic voltammetry (CV) and electrochemical impedance spectroscopy (EIS) in a solution containing ferricyanide, however no redox probe was required for the actual MDPV measurements. The interaction between the poly(*o*-phenylenediamine) imprinted polymer and MDPV was studied by density-functional theory (DFT) methods. The SPCE-MWCNT-AgNP-MIP sensor responded adequately to the variation of MDPV concentration. It was shown that AgNPs enhanced the electrochemical signal by around a 3-fold factor. Making use of square-wave voltammetry (SWV) the developed sensor provided a limit of detection (LOD) of 1.8 μmol L^–1^. The analytical performance of the proposed sensor paves the way to the development of a portable device for MDPV on-site sensing to be applied in forensic and doping analysis.

## 1. Introduction

Methylenedioxypyrovalerone (MDPV), also known as “Magic”, “Vanilla Sky”, “Ivory Wave”, “Super Coke”, or “Energy 1” is one of the many synthetic cathinones intentionally synthesized to be commercialized as a ”legal” alternative to ecstasy (3,4-methylenedioxymethamphetamine, MDMA) [1,2]. MDPV is believed to have entered the drug market in mid-2008, when its consumption spread worldwide, resulting in the emergence of many cases of intoxications and fatal outcomes [3,4,5,6,7]. Like many other new psychoactive substances (NPSs), MDPV is often purchased online as “bath salts”, “legal highs”, “plant food”, or “research chemicals” [8,9]. Since 2012 this substance is classified as a Schedule I drug in the US Controlled Substances Act [10]. The World Anti-Doping Agency (WADA) has later prohibited the use of cathinones and its analogues, such as MDPV, before or during athletic competitions [11]. The European Monitoring Centre for Drugs and Drug Addiction (EMCDDA) attributed the use of MDPV as the cause of dozens of deaths in Europe [12]. According to users’ testimonies, MDPV primarily induces similar positive effects to MDMA, such as euphoria and feeling of increased energy, followed by several undesirable consequences like sleeplessness, memory impairment, agitation, muscle spasms and paranoia [13,14]. In some cases, it can lead to life-threatening side effects like hyperthermia, rhabdomyolysis, seizures, respiratory failure and eventually culminate in fatality [15,16,17,18]. Some clinical studies identified further consequences of MDPV use such as hypertension, chest pain, hyperthermia, persecutory delusions, tachycardia, aggressiveness, and suicidal thoughts [19,20,21,22]. MDPV inhibits the reuptake of norepinephrine, serotonin and, mainly, dopamine, where this substance displays greater potency than amphetamine-like drugs [23]. Although intranasal route seems to be the most established administration route, others include oral, rectal, intravenous, and even inhalation. Depending on the administration route and the purity of the substance, a single dosage usually contains an amount between 5 to 20 mg of MDPV [24]. 

Currently, there are several methods in literature for MDPV quantification (Table 1). They are mainly based on liquid chromatography (LC) associated to a mass spectrometry (MS) detector [25,26,27,28,29,30,31,32,33,34,35,36,37], gas chromatography (GC) associated with MS [38,39,40,41,42,43,44] or flame ionization detection (FID) [41]. There are a few works with other techniques, like ion mobility spectrometry subsequent to liquid-liquid extraction [45], capillary electrophoresis (CE) with in-line solid-phase extraction (SPE) and spectrophotometric detection [46], Raman spectroscopy [47], nuclear magnetic resonance (NMR) spectroscopy [48] and direct MS analysis [49]; there is also a commercially available portable immunoassay [50]. Although most of these methodologies allow the reliable identification of MDPV, they all require some kind of sample pre-treatment [51] and tend to be expensive and time-consuming. In this context, electroanalytical methods could provide more portable, simpler, cost-effective, and faster analysis than aforementioned procedures. However, in the direct analysis of complex samples, electroanalytical methodologies may suffer from severe matrix effects and lack of selectivity. Molecular imprinting technology may be suitable to address such issue [52]. Particularly, electropolymerized molecularly imprinted polymers (e-MIPs) are currently being explored to obtain high performance versatile sensors [53,54,55], which are being applied for a variety of molecules in the medical and forensic fields [56,57,58].

The objective of this work was therefore the development of a selective electrochemical sensor for MDPV based on e-MIPs, using benzene-1,2-diamine, also known as 1,2-phenylenediamine or *ortho*-phenylenediamine (*o*-PD) as the functional monomer and the analyte as the template. The MIP was electropolymerized directly on the surface of a screen-printed carbon electrode (SPCE) previously modified with carboxylated multi-walled carbon nanotubes (MWCNT) and silver nanoparticles (AgNP), and the sensor was applied to the detection of MDPV in biological samples. Although there is a previous, very recent, work where MIPs are combined with AgNPs [59], to the best of the authors’ knowledge none combines MIP-MWCNT with AgNPs, likewise, this is the first time MIPs were developed for MDPV.

## 2. Materials and Methods

### 2.1. Chemicals and Samples

All commercial reagents were of analytical grade and were used without further purification. All aqueous solutions were prepared using ultrapure water with resistivity not less than 18.2 MΩ cm at 298 K (Millipore water purification system, Burlington, MA, USA). Potassium hexacyanoferrate (III), potassium hexacyanoferrate (II) trihydrate, *o*-PD, dopamine, caffeine, tyramine, and potassium chloride (KCl) were obtained from Sigma-Aldrich, St. Louis, MO, USA and amphetamine from Tocris Bioscience, Bristol, UK. Phosphate buffered saline (PBS), 0.17 mol L^–1^, pH 7.4, was prepared using sodium phosphate dibasic and potassium phosphate monobasic, all obtained from Sigma-Aldrich. MDPV hydrochloride was purchased online (www.sensearomatics.eu) and was fully characterized by MS and elemental analysis (shown in the supplementary data of a recent publication [63]). The analytical data were consistent with the expected structure, with the salt having a purity of 99.5%, the salt consisted in a racemic (*R* and *S* enantiomers) mixture.

### 2.2. Equipment

Voltammetric measurements, such as cyclic voltammetry (CV) and square wave voltammetry (SWV), were carried out using an Autolab PGSTAT10 potentiostat/galvanostat controlled by GPES 4.9 software. Electrochemical impedance spectroscopy (EIS) studies were performed using an Autolab PGSTAT204 potentiostat/galvanostat expanded with a FRA32M EIS module (Metrohm, Herisau, Switzerland) and the NOVA 1.10.1.9 software for data acquisition. Fitting of the EIS data was performed with EIS Spectrum Analyser 1.0 [64].

Commercial SPCEs consisting of a carbon working electrode modified with carboxyl functionalized MWCNT (SPCE-MWCNT) with a 4.0 mm diameter, a carbon auxiliary electrode and a silver (Ag) pseudo-reference electrode (DropSens, Llanera, Spain, ref. 110CNT) were used to prepare the electrochemical sensors. 

The electrodes’ surfaces were examined by atomic force microscopy (AFM) using a scanning probe microscope (SPM) operating on atomic force, magnetic force and scanning tunnel microscopy Veeco Metrology Multimode/Nanoscope IVA, and by scanning electron microscopy (SEM) using a high-resolution scanning electron microscope with X-ray microanalysis Quanta 400 FEG ESEM/EDAX Pegasus X4M (FEI, Hillsboro, OR, USA), at the CEMUP Laboratory (University of Porto, Porto, Portugal). 

### 2.3. Preparation of SPCE-MWCNT-AgNP-MIP

The preparation procedure is schematically described in Figure 1. Initially, a commercial SPCE-MWCNT was washed with ultrapure water and dried using a nitrogen flow. Then, a 10 μL aliquot of a 20 mg L^–1^ of 10 nm-AgNP dispersion was carefully drop casted on the surface of the SPCE-MWCNT’s working electrode and dried at 50 °C for 10 min to evaporate the solvent. The fabrication process for the synthesis of the e-MIP consisted of two steps. Firstly, 50 μL of the polymerization solution, containing 0.5 mol L^−1^
*o*-PD (monomer) and 2.5 mol L^–1^ MDPV (template) in PBS, was dropped on the SPCE-MWCNT-AgNP covering the three electrodes, and the electropolymerization was achieved running 5 cyclic voltammograms between −0.2 V and +1.3 V at a scan rate of 50 mV s^−1^. Following polymerization, template molecules were extracted immersing the prepared sensor on a flask containing 6 mL of PBS, under stirring, for 15 min. at room temperature. The sensor was gently washed with water and dried under a nitrogen flow and was ready to be stored or applied to MDPV analysis, through 10 min-incubation with 50 μL of a sample containing MDPV, for the rebinding of the analyte. Its quantification was achieved by directly measuring the current intensity of MDPV oxidation signal by SWV. In parallel, a non-molecularly imprinted polymer (NIP) was constructed as a control, through electropolymerization of the monomer in the absence of the template (polymerization solution containing 0.5 mol L^–1^
*o*-PD in PBS).

### 2.4. Theoretical Studies

The Monte Carlo simulation was performed using the DICE program [65]. Initial parametrization of the MDPV and *o*-PD molecules was made with the use of the LigParGen web server [66,67,68], and dihedral non-bonded interactions were later reparametrized according to the potential energy curves of a rigid scan of their rotations obtained by density-functional theory (DFT) single-point calculations, at the B3LYP [69,70]/6-31G (d,p) [71,72] level of theory and D3 (BJ) dispersion correction [73,74], as implemented in the Gaussian’09 suite [75]. Geometry optimizations were performed with Grimme’s scc-free GFN-xTB method [76] and its GFN2-xTB improvement, which is able to reproduce organic molecule structures with DFT accuracy [77,78]. Final single-point energy calculations were were performed with the ORCA 4.0 package [79]. The ORCA program system [80] and the SMD implicit solvent model, with water as solvent [81], at the at the BP86 [82]/def2-SVP [83] level of theory and RI approximation [84,85] and D3 (BJ) dispersion. The Becke surfaces with electronic density as mapped function were calculated through the quantitative analysis of molecular surface [86] with Multiwfn package [87]. Properties from Bader’s Quantum Theory of Atoms in Molecules (AIM) analysis [77,78] were also calculated with Multiwnf package.

VMD package [88] was used to visualize simulation and render images.

## 3. Results and Discussion

### 3.1. Characterization of the Electrodes during the Modification Process

#### 3.1.1. Electrochemical Evaluation

Electropolymerization was performed in the presence of *o*-PD and MDPV, in an analogous manner to the development of other e-MIPs in literature [89,90,91]. o-PD is largely used to develop MIPs [90,92,93,94,95,96,97,98] due to its highly reactive properties (and ability to polymerize at room temperature, for similar reason is also commonly used as a derivatizing agent [99,100,101]. Interestingly it seems that even though a ‘linear’ poly(*o*-PD) (i.e., when both vicinal amino groups react [102,103,104]) is more stable and produced in larger quantities, it is the ‘ramified’ poly(*o*-PD) (i.e., when only one of the amino group reacts and the other is left free to further reactions [102,105]) that is responsible for the suitable interactions with the template/monomer [90]. During the electrochemical polymerization [106] an irreversible oxidation process appeared during the first cycle and disappeared during the following cycles. The molecularly imprinting polymerization on the electrode surface resulted in a substantial reduction on the charge transfer and the decreasing of the anodic current with the number of CV cycles, suggesting the successful formation of a non-conductive imprinted polymeric film. Two MIP-sensors were subsequently prepared, one directly developed on the MWCNT-carbon surface and the other developed on that surface previously covered with AgNPs. Figure 2A shows that the modification with AgNPs enhanced the sensitivity of the electrode, in the NIP but especially in the MIP, possibly due to the further increase in the effective surface area on which the MIP is obtained, it was not adequately study if the reason was instead possibly electrocatalytic.

As a proof-of-concept of the sensor, a brief SWV study was carried out (Figure 2B). The presence of an oxidation signal corresponding to MDPV in MIP after polymerization proves its presence as template within the polymeric layer. It is worth mentioning that the monomer electrooxidation was performed at lower potentials (ca. 0.25 V) than the template oxidation, avoiding possible drawbacks during electropolymerization [56]. Additionally, MDPV signal disappears following the extraction step, indicating the effectiveness of such process. Indeed, the electrochemical mechanism of MDPV oxidation at a carbon surface has been recently proposed [60]. After incubating the sensor with MDPV, its anodic wave is again observed, confirming the adequate access of this molecule to the sensor surface through the previously formed cavities.

In order to characterize the surface modification of the electrodes’ surface, as well as the resistance of the different modification steps, CV and EIS were performed using 2.5 mmol L^−1^ [Fe (CN)_6_]^3−/4−^ as redox probe. As displayed in Figure 2C, CV of [Fe (CN)_6_]^3−/4−^ at the surface of SPCE-MWCNT the typical well-defined pair of redox peaks is observed due to the high electron transfer rate and the large surface area. Notwithstanding, the current intensity of the redox probe drastically decreases following MIP and NIP polymerization, demonstrating the formation of a non-conductive film on the electrode’s surface. Interestingly, it is suggested that the polymerization occurs more efficiently in the absence of template molecules, which intercalation through the polymeric chain allows greater access of electrons to the electrode surface. This finding is also observed by EIS (Figure 2D), where the semicircle diameter corresponds to the charge transfer resistance that is ascribed to the MIP (R_MIP_) and is higher in the electrode following polymerization of the NIP (9.58 kΩ) than the MIP’s (2.82 kΩ). After the extraction step, in which the MDPV molecules are removed leaving complementary cavities imprinted on the polymeric film, the redox probe obtains easier access to the electrodes surface, resulting in improvement of CV redox signals and decreasing of resistance (R_MIP_ = 0.912 kΩ). When the sensor is incubated with a solution containing the analyte, the rebinding of MDPV molecules slightly changes the conductivity and electron transfer. 

#### 3.1.2. Morphological Study

The surface morphological analysis of the modified electrodes was investigated by AFM, SEM and EDS. The 3-dimensions AFM images are shown in Figure 3. Notwithstanding the fact that the morphology of the unmodified carbon surface is not flat (Figure 3A), which makes it harder to visualize the subsequent changes, it is possible to discern the presence of tubular and spheric structures on the electrode surface after modification with MWCNT and AgNP (Figure 3B). Following formation of the MIP, the electrode topography appears to be less detailed, somehow suggesting that the previously described structures were covered by the polymeric layer (Figure 3C). 

The electrodes’ architecture in different stages of construction was further studied by SEM. As shown in Figure 3D, a typical three-dimensional spaghetti-like structure is observed on the SPCE-MWCNT, whereas on the surface of SPCE-MWCNT-AgNP (Figure 3E), the dispersion of bright spheres with diameters of about 10 nm on the electrode’ surface appears to be uniformly distributed. The composition of the electrode surface at this stage was analyzed by energy-dispersive X-ray spectroscopy (EDS), which revealed the respective peaks of carbon and silver (Figure 3F). Following the molecularly imprinting process, a pronounced modification on the electrode morphology is observed, compatible with the synthesis of a homogenous and compact polymeric layer (Figure 3G), which is preserved after the extraction step, being additionally possible to observe the maintenance of the MWCNT-AgNP structure at the end of the construction procedure (Figure 3H).

### 3.2. Theoretical Studies

It has been long discussed that the formation of the pre-polymerization complex between the analyte and the building monomers in solution is the driving force of the molecular imprinting process [93,107,108,109]. The theoretical study of such a process, for its part, requires a good description of the pre-polymerization complex, but also of the potential energy surface (PES) of the interaction between the analyte and the monomers, which can originate a wide range of different complexes and conformers. In other words, a theoretical study must account for a good sampling of the possible interactions, as much as it needs to describe the monomer-analyte interaction in any given possible compound. In this work, it was sought to achieve this goal by a combination of Monte Carlo sampling simulations and semi-empirical quantum mechanical geometry optimizations and DFT energy calculations, a protocol already presented in the literature in a similar form [110].

After the initial thermalization of a simulation box containing one MDPV molecule at the interface of two layers, one of 601 *o*-PD molecules and another of 5000 water molecules, the production simulation consisted two parallel runs totaling 308 thousand steps using a constant-NPT (Number of particles, Pressure and Temperature) ensemble. Then 616 snapshots of the simulation were used—one for each 500 steps, considering the decorrelation of the simulation steps—to extract all the conformers composed of the MDPV molecule and its first *o*-PD solvation shell. Those 616 pre-polymerization complexes—referred from now on as cavities—then had their geometries optimized with Grimme’s SCC-free GFN-xTB [76] method and separated according to the number of o-PD monomers. Cavities of 16 to 25 monomers were found in the sample, but 92 percent of those cavities had between 18 and 23 monomers. The largest group of cavities (representing 22.7 percent of all) had 20 monomers, while 20.5 percent had 19 monomers and 18.5 had 21 monomers. For each of the most representative groups—from 18 to 23 monomers—the most stable cavities, according to their bonding energies, were chosen for a further geometry optimization with the GFN2-xTB method [111,112]. In this step, a 10 kcal mol^–1^ cut-off was used from the most stable cavity in each group, totaling 46 cavities, which had their local geometry minima confirmed by frequency calculations. Those cavities falling within a 5 kcal mol^−1^ binding energy from the most stable one were finally used for DFT single point energy calculations at the BP86 [82]/def2-SVP [83] level of theory and RI approximation [84] and D3(BJ) dispersion.

The final evaluation considered the most stable cavity within each group of representative cavities. Figure 4 shows that the most stable cavities, by far, are those with 22 and 23 *o*-PD monomers around the MDPV molecule. Their close formation energy (357 against 359.6 kcal mol^–1^) shows that the increased stability is not an arbitrary effect of the aggregation of one more monomer, but of real interactions between the monomers and those and the MDPV molecule. However, the figure also shows that those monomers are representative of only 22 percent of the cavities (7.8 percent in the 23 monomer case), which means they are rather difficult to access from the solution environment, and are likely not responsible for selectivity. One might also expect that once the MDPV molecule had accessed such a hindered, but highly stable cavity, it would not be eluted. On the other hand, 18-monomer cavities are both the less stable and less abundant ones, and not likely to contribute significantly to the selectivity. 

The cavities with 19, 20 and 21 monomers account for over 60 percent of the overall population under those conditions and offer the best insight into the mechanisms of MIP formation. One observes that even with both MDPV and *o*-PD possessing aromatic rings, π-π stacking is not a significant interaction, and bonding occurs almost exclusively between *o*-PD’s amine groups and those MDPV’s methylenedioxy group and carbonyl group. One also may realize that MDPV’s tertiary amine is rather hindered while the MDPV molecule interacts with an *o*-PD shell—a key conclusion driven by Monte Carlo sampling—to be able to contribute to the cavities’ selectivity. 

To evaluate the distribution of non-covalent interactions between *o*-PD monomers and MDPV in the MIP pre-polymerization complexes, analysis of Becke surface [113] was performed for the pre-polymerization cavities with higher representativity, i.e., the complexes with 19, 20 and 21 monomers. This analysis was based on Hirshfeld surface [114], applying the Becke’s [115], instead of the Hirschfeld’s, partition. Moreover, to further investigate the non-covalent interactions, AIM analysis [77,78] was performed on the three most representative cavities, formed with 19, 20 and 21 monomers (20.5%, 22.7% and 18.5%, respectively). Each analysis was calculated from their respective wavefunction file calculated with the BP86 [82] functional and def2-SVP [83] basis set with water as solvent through the implicit solvent model SMD [81], by applying the Orca package [79].

The Becke surfaces (Appendix A) demonstrate that the main non-covalent interactions are formed between O atoms from MDPV and *o*-PD on pre-polymerization complexes. In complexes formed with 19 and 20 monomers, the N atom from the pyrrolidine group of MDPV also showed to establish non-covalent interactions. Thus, AIM analysis of pre-polymerization complexes was performed to further evaluate these interactions. AIM properties of interactions between O or N atoms from MDPV and *o*-PD monomers are summarized in Appendix A.

Through the analysis of the AIM molecular graph of the most representative pre-polymerization complex (with 20 *o*-PD monomers), showed in Figure 5, several Bond Paths (BPs)—which connected two attractors—and its respective Bond Critical Points (BCPs) between atoms from MDVP and from *o*-PD monomers can be identified. Appendix A show the graphs of the pre-polymerization complexes with 19 and 21 *o*-PD monomers, respectively. Through AIM analysis it was demonstrated that the analyte (MDPV) interacts through several non-covalent interactions with *o*-PD monomers to form pre-polymerization cavities. Furthermore, AIM properties of the interactions through O and N atoms of MDPV and *o*-PD monomers demonstrate that all these interactions are non-covalent, featured by their positive values of Laplacian electronic density, ∇^2^ρ(r), and values of density of total energy, *H*(r), close to zero. Besides that, the Binding Energies (*BE*) of the non-covalent interactions calculated through the equation of Espinosa (Equation (1)) demonstrate that these interactions are weak. However, the high number of interactions stabilizes the analyte in the pre-polymerization cavities and make the cavities selective to MDPV.
*BE* = *V*(r)/2(1)
where *V*(r) is the density of potential energy.

Through this analysis, it was demonstrated that the similar affinity for those compounds is indicated to be directly tied to the main driving forces in the formation of the pre-polymerization complex: hydrogen bonding to the benzocaine’s amine group and π-π stacking between benzocaine’s aromatic system and the 3,4-AHBA monomers. In fact, hydroxyzine is the compound with the largest affinity for the polymer, and it has a chlorine replacing the amine group playing a double role: while it is able to accept hydrogen bonds from the polymer, it is also able to direct electron density towards the aromatic system, contributing to its interaction with the stacked monomers. On the other hand, the aminopyrine molecule is the fourth with the largest affinity, just below benzocaine itself. It has no group analogue to benzocaine’s amine but features a pyrazolone attached to the benzene ring that is able to interact with it by hyperconjugation, maintaining a planarity similar to the carboxyl-benzene system in benzocaine and thus also allowing for π-π stacking with the 3,4-AHBA monomers.

### 3.3. Influence of the Experimental Conditions

Studies were conducted regarding the effect of several experimental parameters on sensor’s performance, including the concentration of monomer (*o*-PD) and template (MDPV), number of electropolymerization cycles and period of incubation for MDPV rebinding in the specific cavities formed during MIP development. Measurements were performed through SWV analysis of a 1.5 mmol L^−1^ MDPV solution; please note that the results are presented as the difference between the peak current intensity (i_p_) of the MIP and the i_p_ of the NIP, Δi_p_. 

The ratio monomer:template during electropolymerization process is essential to successfully obtain a MIP, influencing its stability and selectivity to the target molecule. Several polymerization solutions were tested, containing different amounts of *o*-PD, ranging from 0.25 to 10 mmol L^−1^, and MDPV, in the range between 0.25 and 7.5 mmol L^−1^. The sensor prepared using 0.5 and 2.5 mmol L^−1^ of o-PD and template, respectively, exhibited the best analytical performance and was therefore selected to the following experiences, representing a monomer:template ratio of 1:5. The number of polymerization cycles significantly influenced MDPV peak current. The optimum number of cycles to was 5, other tested number of cycles (namely 2, 10, 15 and 20) were found to considerable decrease the sensor sensibility. The higher the number of polymerization cycles the thicker the obtained layer and therefore the harder it is to successfully extract the template molecules and form specific cavities. If the layer is however too thin it can be rather unstable or incapable of properly polymerize around template molecules. In order to choose the best incubation time, the analysis of MDPV was performed following incubation during 1, 5, 10, 20 and 60 min. The sensibility towards MDPV gradually increased with the incubation period, being maximum after 10 min.

The calculated film thickness (*h*) was calculated to be 13 ± 2 nm, using the following formula [52]:*h* = *qM*/*ρAnF*(2)
where *q* is the charge associated with the polymerization, obtained by integrated the peaks during electropolymerization, *M* is the o-PD’s molecular weight, *ρ* is the polymer density (a value of 1.2 g cm^−3^ was assumed), *A* is the electrode geometric area, *n* is the number of electrons involved in the electropolymerization of each *o*-PD molecule, and *F* is the Faraday constant. This low thickness value is consistent with the fact that poly(*o*-PD) is not a good conducting material, so the molecules of MPDV cannot be too far away from the SPCE-MWCNT-AgNP-MIP surface.

### 3.4. Analytical Performance

To evaluate the sensitivity of the sensor, its analytical performance towards the oxidation of MDPV was studied by SWV under optimal conditions. As shown in Figure 6, the SWV current response increased with the increase of MDPV concentration. Moreover, two MIP-sensors were tested, the one directly built on the surface of MWCNT and the one constructed on the AgNPs-MWCNTs. Their respective linear regression equations were: i_p_ (in A) = (264 ± 6) × 10^−4^ [MDPV] (in mol L^−1^) + (57 ± 5) × 10^−8^, r^2^ = 0.9986 (*n* = 5); and i_p_ (in A) = (567 ± 6) × 10^−4^ [MDPV] (in mol L^−1^) + (1 ± 2) × 10^−8^, r^2^ = 0.9997 (*n* = 5). Although both sensors’ response exhibited a good linear relationship between anodic current and MDPV levels, the SPCE-MWCNTs-AgNP-MIP provided a lower limit of detection (LOD) of 1.8 μmol L^−1^ (vs. 6.3 μmol L^−1^ of SPCE-MWCNTs-MIP) and quantification (LOQ) of 6.1 μmol L^−1^ (vs. 21 μmol L^−1^ of SPCE-MWCNTs-MIP), therefore satisfying the needs for MDPV analysis in forensic samples. When compared to other MDPV analytical methodologies (Table 1), the proposed method exploited the use of an e-MIP to mimic an antigen-antibody bond and turn the procedure quite selective beyond sensitive. In addition, this sensor is a cheaper and more practical alternative to the previously described methods for MDPV analysis, with the advantage of being constructed in a quickly, simple, and inexpensive way.

Moreover, a 10 μmol L^−1^ MDPV solution was repeatedly evaluated resulting in a repeatability of 6% (*n* = 5). The selectivity of the sensor was assessed comparing the MDPV signal (i_p_ = 15.2 μA, E_p_ = 0.780 V) with several molecules important in forensic sciences (all compounds in a concentration of 1.5 mmol L^−1^): caffeine (non-detected), MDMA (i_p_ = 20.0 μA, E_p_ = 1.068 V), amphetamine (non-detected), methamphetamine (non-detected), dopamine (i_p_ = 26.8 μA, E_p_ = 0.206 V), and tyramine (i_p_ = 23.4 μA, E_p_ = 0.554 V). A preliminary analysis was performed in blood serum (Appendix A).

Reports of MDPV intoxication cases in concentrations ranging from 2.0 × 10^7^ to 4.8 × 10^9^ ng mL^−1^ in blood, and from 7.6 × 10^2^ to 1.4 × 10^3^ ng mL^−1^ in urine, were reported [12], being the performance of the SPCE-MWCNT-AgNP-MIP sensor appropriate for its medical and forensic application. Although the LOD is not so low as in other published works (Table 1), those methodologies are mostly non electroanalytical but mainly based in a chromatographic separation, which makes this alternative particularly advantageous in terms of portability and speed. It is worth mentioning that this study was performed using a racemic mixture, however recent studies are showing that the *S* enantiomer is more potent than the *R* [116], thus this is something that should be addressed in future electroanalytical studies including in the theoretical simulation step [117,118].

## 4. Conclusions

An electrochemical SPCE-MWCNT-AgNP-MIP sensor was developed and successfully applied to the analysis of MDPV. In addition to the speed and simplicity, the e-MIP proved to be sufficiently resistant and robust, and the formed cavities demonstrated adequate selectivity for the target molecule, that had also served as the template. By combining MWCNT and AgNPs, the sensor achieved a LOD of 1.8 μmol L^−1^, repeatability of 6%, along with suitably selectivity. The experimental results confirmed that the sensor has the capacity of detecting MDPV in clinically relevant concentrations. Its successful application to biological samples will potentially make it a suitable alternative in forensic analysis. 

## Figures and Tables

**Figure 1 nanomaterials-11-00353-f001:**
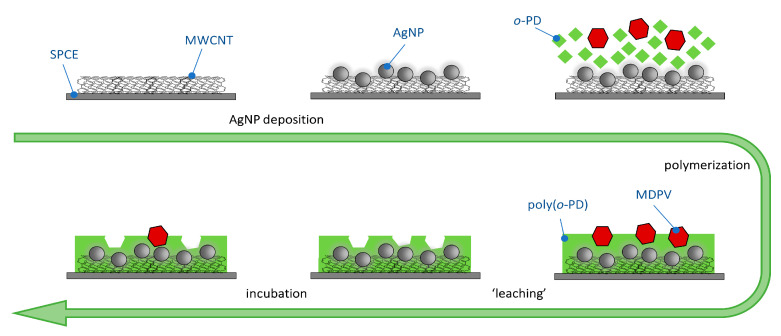
Schematics of the development of the SPCE-MWCNT-AgNP-MIP sensor.

**Figure 2 nanomaterials-11-00353-f002:**
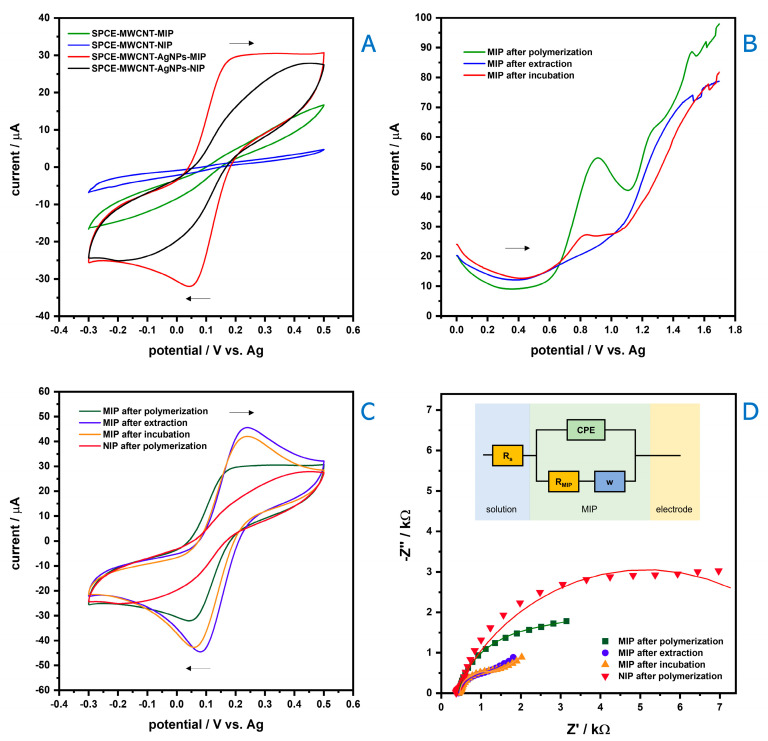
CV characterization of MIP and NIP sensors developed on SPCE-MWCNT and SPCE-MWCNT/AgNP electrodes (**A**); SWVs of a 0.1 mol L^−1^ MDPV solution after polymerization, extraction and incubation steps (**B**); Characterization of the step-by-step construction of the SPCE-MWCNT/AgNP-MIP sensor through CV (**C**) and EIS Nyquist diagrams with suitable fitting (the circuit used is shown within the figure) (**D**). Measurements (**A**,**C**,**D**) were performed using a 50 μL PBS solution containing 2.5 mmol L^–1^ [Fe (CN)_6_)]^3−/4−^ and 0.1 mol L^–1^ KCl.

**Figure 3 nanomaterials-11-00353-f003:**
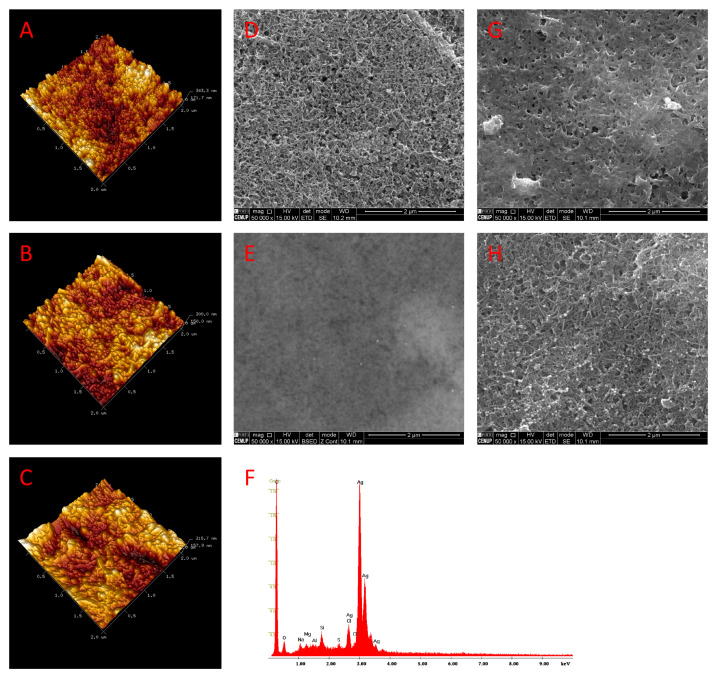
AFM images of bare-SPCE (**A**), SPCE-MWCNT-AgNP (**B**), and SPCE-MWCNT-AgNP-MIP (**C**); SEM images (50,000 magnification) of SPCE-MWCNT (**D**), SPCE-MWCNT-AgNP (**E**), SPCE-MWCNT-AgNP-MIP after polymerization (**G**), and SPCE-MWCNT-AgNP-MIP after extraction (**H**); EDS analysis of SPCE-MWCNT-AgNP (**F**).

**Figure 4 nanomaterials-11-00353-f004:**
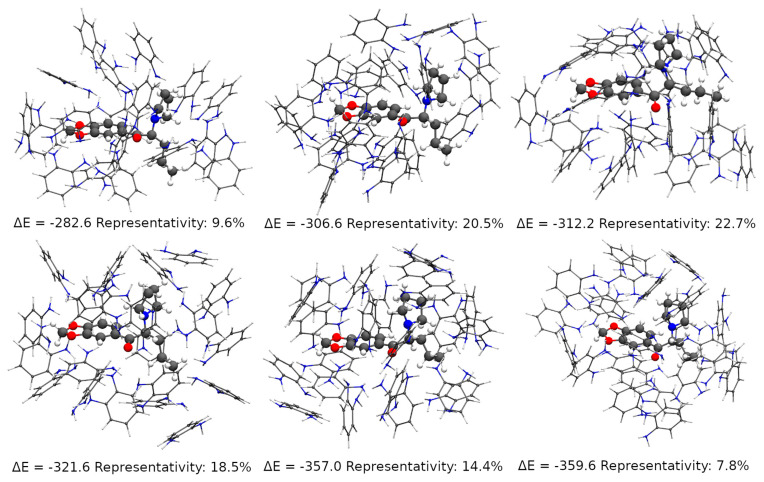
Most representative and stable cavities formed by *o*-PD monomers around MDPV. Energies in kcal mol^−1^.

**Figure 5 nanomaterials-11-00353-f005:**
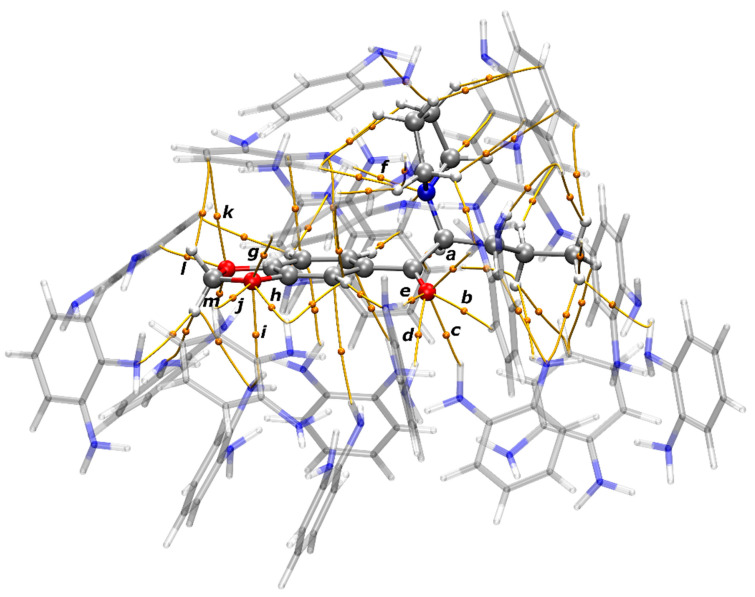
AIM molecular graph of the MIP pre-polymerization cavity with MDVP and 20 *o*-PD monomers. Orange lines are the Bond Paths (BP) that connect two attractors, and the orange spheres are the Bond Critical Points (BCP).

**Figure 6 nanomaterials-11-00353-f006:**
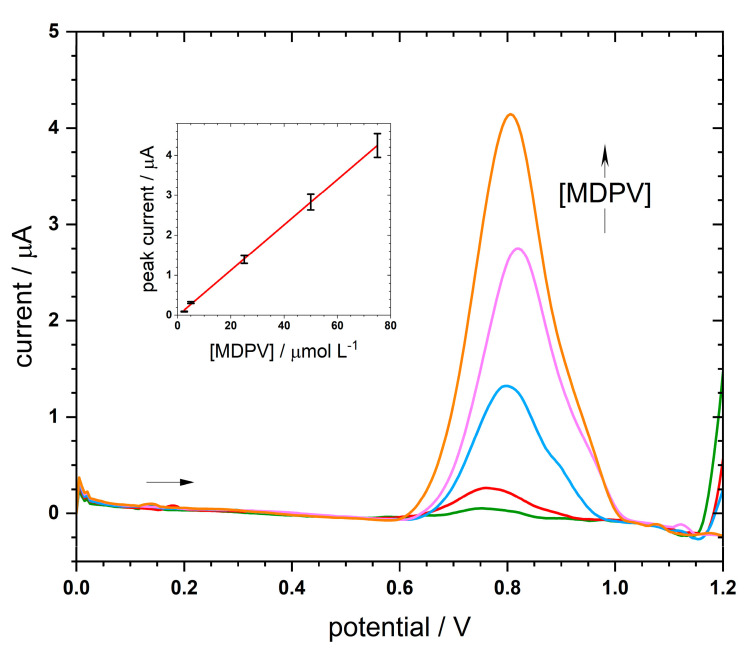
SWVs of different MDPV solutions, inlay the corresponding calibration curve.

**Table 1 nanomaterials-11-00353-t001:** Analytical methods reported for MDPV.

Technique	Extraction	LOD/ng mL^−1^	LOQ/ng mL^−1^	Matrix	Reference
SWV	none	5 × 10^2^	2 × 10^3^	buffer	this work
AdSDPV	none	2 × 10^2^	5 × 10^2^	seized samples	[60]
LC-MS	LLE	2	10	seized samples	[25]
LC-MS	SPE	2 × 10^−4^	1 × 10^−3^	wastewater	[27]
LC-MS	SPE	1	5	rat brain tissue	[29]
LC-MS	SALLE	2	4	urine	[31]
LC-MS	PHPP	0.1	0.25	plasma	[32]
LC-MS	LLE	2 × 10^−3^	5 × 10^−3^	equine plasma	[34]
LC-MS	SPE	-	5 × 10^−3^	wastewater	[35]
LC-MS	LLE	0.5	5	blood	[36]
LC-MS	SPE	3 × 10^−2^	0.5	saliva	[37]
GC-MS	LLE	7	2 × 10^1^	blood, urine	[61]
GC-MS	LLE and derivatization	-	2 × 10^1^	urine	[39]
GC-MS	SPE and derivatization	2 × 10^1^	5 × 10^1^	urine	[42]
GC-MS, LC-MS	SPE	2 × 10^3^	-	hair, kidney, liver, bile	[44]
IMS	LLME	2 × 10^1^	7 × 10^1^	oral and nasal fluid	[45]
CE-UV	PLE-SPE	1 × 10^5^	4 × 10^5^	hair	[46]
CE-MS	SPE	1 × 10^1^	3 × 10^1^	urine	[62]
IM-MS	-	1 × 10^4^	-	standards	[49]
immunoassay	none	0.2	-	urine	[50]

AdSDPV: adsorptive stripping differential pulse voltammetry; CE-UV: capillary electrophoresis with ultra-violet detection; GC: gas chromatography; IMS: ion-mobility spectroscopy; LC: liquid chromatography; LLE: liquid-liquid extraction; LOD: limit of detection; LOQ: limit of quantification; MS: mass spectroscopy; PHPP: plasma hydrolysis and protein precipitation; PLE: pressurized liquid extraction; SALLE: salting-out liquid-liquid extraction; SPE: solid-phase extraction.

## Data Availability

The data presented in this study are available on request from the corresponding author.

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
