# Peer review of "3,4-Methylenedioxypyrovalerone (MDPV) Sensing Based on Electropolymerized Molecularly Imprinted Polymers on Silver Nanoparticles and Carboxylated Multi-Walled Carbon Nanotubes"

_nanomaterials, 2021, doi:10.3390/nano11020353_

Round 1
Reviewer 1 Report
Review: nanomaterials-1067548.
Title: 3,4-methylenedioxypyrovalerone (MDPV) sensing based on electropolymerized molecularly imprinted polymers on silver nanoparticles and carboxylated multi-walled carbon nanotubes.
In this research-type manuscript, the sensor for 3,4-methylenedioxypyrovalerone was proposed based on the electropolymerization of 1,2-phenylenediamine on the screen-printed carbon electrode modified with silver nanoparticle and multi-walled carbon nanotubes. The characterization of the electrode was carried out in the terms of cyclic voltammetry and square wave voltammetry, atomic force microscopy and X-ray electron dispersive spectroscopy. The molecular modeling was also employed and brief analytical performance was presented. The manuscript has a potential since the selective materials dedicated to analysis of illicit psychoactive compounds are important from the forensic and toxicological points of view. However, the analysis of manuscript revealed several serious drawbacks that are pointed below:
- The behavior of sensor should be clearly demonstrated by Authors with respect to the sensor designed previously towards 3,4-methylenedioxymethcathinone (see: Sensors Actuators B Chem. 2020, 316, 128133). The comparison of analytical performances of both sensors towards both compounds, viz. 3,4-methylenedioxymethcathinone and 3,4-methylenedioxypyrovalerone could be essential to prove the novelty and applicability. It will be interesting to disclose specificity of 3,4-methylenedioxypyrovalerone sensor towards 3,4-methylenedioxymethcathinone since the specificity of 3,4-methylenedioxymethcathinone sensor towards 3,4-methylenedioxypyrovalerone was low (see Fig. 5 in Sensors Actuators B Chem. 2020, 316, 128133). The mechanism of molecular recognition shall be discussed.
- Template molecule of 3,4-methylenedioxypyrovalerone possess one stereogenic center at the C2 in aliphatic chain. Please, specify, if the racemic mixture was used as the template or any specific enantiomer was applied. If the racemic mixture was used in the synthesis, due to chiral carbon at the template molecule, please precise the theoretical analysis description. Does the model consisted of 50% of the optimized molecules of R-enantiomers and 50% of S-enantiomers? Does the energies calculated were a mean of both values or were similar for both enantiomers? (see: Journal of Materials Science 2020, 55, 10626 and Journal of Molecular Recognition 2018, 31, e2612). As far as it could be seen in Fig. 4, only S-enantiomer was presented. Please, explain the reason?
- Description of molecular modeling should be more precise. It is difficult to agree with Authors that the cavity was created in silico since the molecules of 1,2-phenylenediamine monomer were not covalently bound together mimicking polymerization step (see: Analyst 2014, 139, 1779). In my opinion, at this stage of modeling, it should be considered only as the prepolymerization complex. Please discuss in details the interactions between the template and monomer.
- In order to increase the scientific value of the manuscript, it could be interesting to build a cavity from monomers and discuss the binding energies of various interfering compound, for instance those presented in Fig. 5 in Sensors Actuators B Chem. 2020, 316, 128133. In silico analysis of interactions in the cavity could help to discuss the mechanism of molecular recognition.
- In the Introduction Section, in order to reveal full potential of imprinted materials, the recent reviews devoted to imprinting technology and their widespread application shall be described with proper references (see: Chemical Reviews 2019, 119, 94, Chemical Society Reviews 2016, 45, 2137, Chemical Reviews 2020, 120, 9554).
- Please, explain the mechanism of 1,2-phenylenediamine polymerization and the structure of polymer. Please, refer to literature. Additionally, please use IUPAC nomenclature (according to recent guidelines, o-phenylenediamine is not preferred name for 1,2-phenylenediamine).
- Please discuss the oxidation potentials of template and monomer since it could generate problems during electropolymerization (see: Biosensor Bioelectronics 2020, 169, 112589). The reference to Electrochimica Acta 2020, 354, 136728 could be insufficient.
- In my opinion, the analytical part of the manuscript is insufficient. The real sample analysis should be completed together with discussion of the selectivity with respect to oxidation potential of interfering compounds that can occur in real sample environment (please note that the analyte undergoes the first pass metabolism).
- In Results and Discussion Section (line 183-184) it was stated that silver nanoparticles enhance electrochemical signal probably due to the further increase in the effective surface area. Please extend the discussion and support the hypothesis by data. Does the silver nanoparticles formed a thin layer? What kind of interactions allowed to conjugate the MIP layer on the surface of silver nanoparticles? Was the MIP layer durable on silver nanoparticles support?
- The last sentence of Conclusions is very optimistic and it should be proved by data from real sample analysis.
In my opinion, the manuscript has some potential but serious revision is needed at this stage of evaluation.
Based on above, I recommend major revision of the manuscript.
Author Response
In this research-type manuscript, the sensor for 3,4-methylenedioxypyrovalerone was proposed based on the electropolymerization of 1,2-phenylenediamine on the screen-printed carbon electrode modified with silver nanoparticle and multi-walled carbon nanotubes. The characterization of the electrode was carried out in the terms of cyclic voltammetry and square wave voltammetry, atomic force microscopy and X-ray electron dispersive spectroscopy. The molecular modeling was also employed and brief analytical performance was presented. The manuscript has a potential since the selective materials dedicated to analysis of illicit psychoactive compounds are important from the forensic and toxicological points of view.
First, we would like to thank the reviewer for taking the time to evaluate our manuscript. Thanks!
However, the analysis of manuscript revealed several serious drawbacks that are pointed below:
1. The behavior of sensor should be clearly demonstrated by Authors with respect to the sensor designed previously towards 3,4-methylenedioxymethcathinone (see: Sensors Actuators B Chem. 2020, 316, 128133). The comparison of analytical performances of both sensors towards both compounds, viz. 3,4-methylenedioxymethcathinone and 3,4-methylenedioxypyrovalerone could be essential to prove the novelty and applicability. It will be interesting to disclose specificity of 3,4-methylenedioxypyrovalerone sensor towards 3,4-methylenedioxymethcathinone since the specificity of 3,4-methylenedioxymethcathinone sensor towards 3,4-methylenedioxypyrovalerone was low (see Fig. 5 in Sensors Actuators B Chem. 2020, 316, 128133). The mechanism of molecular recognition shall be discussed.
Unlike MDMA and methylone, MDPV has a pyrrole group, which can determine additional steric interference and largely justify its interference in the methylone sensor that acted by indirectly analyzing the analyte concentration through the use of a redox probe. The reverse phenomenon is far less likely to happen. Still, we tested MDMA, which has a chemical structure very similar to MDPV.
2. Template molecule of 3,4-methylenedioxypyrovalerone possess one stereogenic center at the C2 in aliphatic chain. Please, specify, if the racemic mixture was used as the template or any specific enantiomer was applied. If the racemic mixture was used in the synthesis, due to chiral carbon at the template molecule, please precise the theoretical analysis description. Does the model consisted of 50% of the optimized molecules of R-enantiomers and 50% of S-enantiomers? Does the energies calculated were a mean of both values or were similar for both enantiomers? (see: Journal of Materials Science 2020, 55, 10626 and Journal of Molecular Recognition 2018, 31, e2612). As far as it could be seen in Fig. 4, only S-enantiomer was presented. Please, explain the reason?
A racemic mixture was used in the synthesis, however, it was not detected enantioselectivity for any of enantiomers (R or S). Therefore, in the computational study it was decided to calculate the pre-polymerization complexes only for one of the enantiomers, in this case, the S-enantiomer. Moreover, we now include a final sentence in the Results and Discussion section, advising scientists to take that in mind in future theoretical studies. We also now cite the mentioned manuscripts.
3. Description of molecular modeling should be more precise. It is difficult to agree with Authors that the cavity was created in silico since the molecules of 1,2-phenylenediamine monomer were not covalently bound together mimicking polymerization step (see: Analyst 2014, 139, 1779). In my opinion, at this stage of modeling, it should be considered only as the prepolymerization complex. Please discuss in details the interactions between the template and monomer.
Only the pre-polymerization complexes were used in our computational study. However, as we mentioned in the second paragraph of the “Theoretical studies” section, the pre-polymerization complexes were called cavities in order to make the discussion simpler. In the manuscript we added: “Those 616 pre-polymerization complexes are referred from now on as cavities”). About the second question, to further discuss the interactions between the analyte (MDPV) and the monomers (o-PD) we have performed a Becke surface’s analysis and AIM analysis of the interactions between the analyte and the associated monomers. The analysis and discussion were added to the manuscript, as well as the suggested reference.
4. In order to increase the scientific value of the manuscript, it could be interesting to build a cavity from monomers and discuss the binding energies of various interfering compound, for instance those presented in Fig. 5 in Sensors Actuators B Chem. 2020, 316, 128133. In silico analysis of interactions in the cavity could help to discuss the mechanism of molecular recognition.
Becke surface’s analysis and an AIM study were added to the computational section in order to evaluate the intermolecular interactions in the pre-polymerization complexes. Besides that, we have calculated the binding energies for the main non-covalent interactions between the analyte and the monomers around. The analysis and respective discussion were added to the manuscript.
5. In the Introduction Section, in order to reveal full potential of imprinted materials, the recent reviews devoted to imprinting technology and their widespread application shall be described with proper references (see: Chemical Reviews 2019, 119, 94, Chemical Society Reviews 2016, 45, 2137, Chemical Reviews 2020, 120, 9554).
We thank the referee for the suggestion. We improved this section of the manuscript as proposed and included the adequate references.
6. Please, explain the mechanism of 1,2-phenylenediamine polymerization and the structure of polymer. Please, refer to literature. Additionally, please use IUPAC nomenclature (according to recent guidelines, o-phenylenediamine is not preferred name for 1,2-phenylenediamine).
More about 1,2-phenylenediamine polymerization was added, including several references.
We have corrected it the nomenclature, thank you for the suggestion.
7. Please discuss the oxidation potentials of template and monomer since it could generate problems during electropolymerization (see: Biosensor Bioelectronics 2020, 169, 112589). The reference to Electrochimica Acta 2020, 354, 136728 could be insufficient.
We have improved the discussion, thank you for the suggestion.
8. In my opinion, the analytical part of the manuscript is insufficient. The real sample analysis should be completed together with discussion of the selectivity with respect to oxidation potential of interfering compounds that can occur in real sample environment (please note that the analyte undergoes the first pass metabolism).
Analyzing drug samples is always logistically complicated, 2020 had the added complication of COVID-19, for these reasons the added value of asking the police for real drug samples with analyte was considered not worth it in terms of benefit-cost ratio for the present study.
About the ‘first-pass effect’ that matters to analyze biological fluids, it is no so relevant if one whishes to analyze non-consumed products.
9. In Results and Discussion Section (line 183-184) it was stated that silver nanoparticles enhance electrochemical signal probably due to the further increase in the effective surface area. Please extend the discussion and support the hypothesis by data. Does the silver nanoparticles formed a thin layer? What kind of interactions allowed to conjugate the MIP layer on the surface of silver nanoparticles? Was the MIP layer durable on silver nanoparticles support?
We were careful to say ‘possibly’ instead of ‘probably’, we are not sure if it is not electrocatalytic, further studies would be required. Still, we have improved the text. Thank you for the suggestion. Also, SEM images suggest silver nanoparticles are homogeneously distributed, possibly in a monolayer, but additional techniques would be necessary to confirm it and to deeply study the way they interact with the MIP.
10. The last sentence of Conclusions is very optimistic and it should be proved by data from real sample analysis.
The referee is right, we have changed this sentence. Thank you for the suggestion.
In my opinion, the manuscript has some potential but serious revision is needed at this stage of evaluation.
Based on above, I recommend major revision of the manuscript.
Reviewer 2 Report
In this manuscript, an electrochemical SPCE-MWCNT-AgNP-MIP sensor was developed and successfully applied to the analysis of MDPV. The work reports some reasonable results, which are promising, however, it is not well organized and there are still some issues need to be addressed. Therefore, I suggest major suggestion.
- How about the sensing performance in real world matrix such as blood or other biofluids?
- Characterization of the sensor selectivity to MDPV over non-target molecules is missing.
- According to Table 1, the sensor LOD is 500ng/mL,which is not competitive with other reported MDPV detection methods, so the advantages and significance of the sensor is confused.
- In the inset graph of Figure 5, the signal error bar at each concentration should be added.
- In the line 325, what is the meaning of the symbol ”?” in the equation.
Author Response
In this manuscript, an electrochemical SPCE-MWCNT-AgNP-MIP sensor was developed and successfully applied to the analysis of MDPV. The work reports some reasonable results, which are promising, however, it is not well organized and there are still some issues need to be addressed. Therefore, I suggest major suggestion.
First, we would like to thank the reviewer for taking the time to evaluate our manuscript. Thanks!
1. How about the sensing performance in real world matrix such as blood or other biofluids?
Analyzing drug samples is always logistically complicated, 2020 had the added complication of COVID-19, for these reasons the added value of asking the police for real drug samples with analyte was considered not worth it in terms of benefit-cost ratio for the present study.
2. Characterization of the sensor selectivity to MDPV over non-target molecules is missing.
Although the information is not shown in a table it is shown in a paragraph in page 12.
3. According to Table 1, the sensor LOD is 500 ng/mL, which is not competitive with other reported MDPV detection methods, so the advantages and significance of the sensor is confused.
For example, the MIP sensor is way more portable than LC-MS. The advantages are better described now. Thank you for the suggestion.
4. In the inset graph of Figure 5, the signal error bar at each concentration should be added.
Corrected as suggested. Thank you for the suggestion.
5. In the line 325, what is the meaning of the symbol ”?” in the equation.
It was a “ro”, symbol of density. The problem was on the conversion to the pdf. We hope it is now corrected.
Reviewer 3 Report
I found this manuscript of very good quality; however, I have a single suggestion for possible, minor revision.
In the manuscript reference list, three papers (ref. 40, 43 and 47) focus on the exixtence of MDPVI regioisomers. Therefore, Authors are aware about that. Indeed, the MDPV formulation in bath salts is a racemic mixture, and the S isomer is much more potent than the R isomer at blocking dopamine transporters and producing abuse-related effects (see Baumann MH et al. Neuropharmacology of 3,4-Methylenedioxypyrovalerone (MDPV), Its Metabolites, and Related Analogs. Curr Top Behav Neurosci. 2017;32:93-117. doi: 10.1007/7854_2016_53. PMID: 27830575; PMCID: PMC5392131).
My curiosity (and subsequent suggestion) is: did the Authors consider detection specificity of their sensor with respect to MDPV isomers? If their response is no and this cannot be addressed in this work, it would be nice at least to discuss the point, as a perspective development, in the manuscript.
Author Response
I found this manuscript of very good quality; however, I have a single suggestion for possible, minor revision.
First, we would like to thank the reviewer for taking the time to evaluate our manuscript. Thanks!
In the manuscript reference list, three papers (ref. 40, 43 and 47) focus on the exixtence of MDPVI regioisomers. Therefore, Authors are aware about that. Indeed, the MDPV formulation in bath salts is a racemic mixture, and the S isomer is much more potent than the R isomer at blocking dopamine transporters and producing abuse-related effects (see Baumann MH et al. Neuropharmacology of 3,4-Methylenedioxypyrovalerone (MDPV), Its Metabolites, and Related Analogs. Curr Top Behav Neurosci. 2017;32:93-117. doi: 10.1007/7854_2016_53. PMID: 27830575; PMCID: PMC5392131).
My curiosity (and subsequent suggestion) is: did the Authors consider detection specificity of their sensor with respect to MDPV isomers? If their response is no and this cannot be addressed in this work, it would be nice at least to discuss the point, as a perspective development, in the manuscript.
We used a racemic mixture. When we started the study, we were not so aware of the differences. However, and following the referee’s suggestion, we now include a final sentence in the Results and Discussion section advising scientists to take that in mind in future analytical studies. We also now cite the mentioned manuscript.
Many thanks for the suggestion!
Round 2
Reviewer 1 Report
Review: nanomaterials-1067548.
Title: 3,4-methylenedioxypyrovalerone (MDPV) sensing based on electropolymerized molecularly imprinted polymers on silver nanoparticles and carboxylated multi-walled carbon nanotubes.
In this revised manuscript, the Authors have made corrections according to referee comments. In my opinion, the manuscript in current form could be considered for acceptance.
Author Response
Thank you for your time.
Reviewer 2 Report
This paper can be accepted in present form
Author Response
Thank you for your time.